# Epidemiology of Hereditary Diseases in the Karachay-Cherkess Republic

**DOI:** 10.3390/ijms21010325

**Published:** 2020-01-03

**Authors:** Rena A. Zinchenko, Amin Kh. Makaov, Andrey V. Marakhonov, Varvara A. Galkina, Vitaly V. Kadyshev, Galina I. El’chinova, Elena L. Dadali, Lyudmila K. Mikhailova, Nika V. Petrova, Nina E. Petrina, Tatyana A. Vasilyeva, Polina Gundorova, Alexander V. Polyakov, Oksana Y. Alexandrova, Sergey I. Kutsev, Eugeny K. Ginter

**Affiliations:** 1Research Centre for Medical Genetics, 115522 Moscow, Russia; renazinchenko@mail.ru (R.A.Z.); vgalka06@rambler.ru (V.A.G.); vvh.kad@gmail.com (V.V.K.); elchinova@med-gen.ru (G.I.E.); genclinic@yandex.ru (E.L.D.); npetrova63@mail.ru (N.V.P.); petrina_nina@bk.ru (N.E.P.); vasilyeva_debrie@mail.ru (T.A.V.); p_gundorova@inbox.ru (P.G.); polyakov@med-gen.ru (A.V.P.); kutsev@mail.ru (S.I.K.); ekginter@mail.ru (E.K.G.); 2N.A. Semashko National Research Institute of Public Health, 105064 Moscow, Russia; 3Municipal Budgetary Health Care Setting “Khabez Central District Hospital”, 369400 Khabez, Russia; makaov@yandex.ru; 4N.N. Priorov Central Research Institute of Traumatology and Orthopedics, 117997 Moscow, Russia; cito-uchsovet1@mail.ru; 5Moscow Regional Research and Clinical Institute (“MONIKI”), 129110 Moscow, Russia; o.aleksandrova@monikiweb.ru

**Keywords:** genetic epidemiology, diversity of hereditary diseases, genetic load, Karachay-Cherkess Republic

## Abstract

Prevalence and allelic heterogeneity of hereditary diseases (HDs) could vary significantly in different human populations. Current knowledge of HDs distribution in populations is generally limited to either European data or analyses of isolated populations which were performed several decades ago. Thus, an acknowledgement of the HDs prevalence in different modern open populations is important. The study presents the results of a genetic epidemiological study of hereditary diseases (HDs) in the population of the Karachay-Cherkess Republic (KChR). Clinical screening of a population of 410,367 people for the identification of HDs was conducted. The population surveyed is represented by five major ethnic groups—Karachays, Russians, Circassians, Abazins, Nogais. The study of the populations was carried out in accordance with the proprietary protocol of genetic epidemiological examination designed to identify >3500 HDs easily diagnosed during clinical examination by qualified specialists specializing in the HDs. The protocol consists of the population genetic and medical genetic sections and is intended for comprehensive population analysis based on the data on different genetic systems, including the genes of HDs, DNA polymorphisms, demographic data collected during hospital-based survey. 8950 families (with 10,125 patients) with presumably the HDs were initially identified as a result of the survey and data collection through various sources of registration (from 1156 medical workers from 163 medical institutions). A diagnosis of hereditary pathology was established in 1849 patients (from 1295 families). Two hundred and thirty nosological forms were revealed (in 1857 patients from 1295 families). The total prevalence of HDs was 1:221. Differences between populations and ethnic groups were identified: 1:350 in Russians, 1:195 in Karachays, 1:199 in Circassians, 1:218 in Abazins, 1:135 in Nogais. Frequent diseases were determined, the presence of marked genetic heterogeneity was identified during the confirmatory DNA diagnosis. To explain the reasons for the differentiation of populations by load of HD, a correlation analysis was carried out between the F_ST_ (random inbreeding) in populations and HDs load values. This analysis showed genetic drift is probably one of the leading factors determining the differentiation of KChR populations by HDs load. For the first time, the size of the load and spectrum of HDs in the populations of the KChR are determined. We have demonstrated genetic drift to be one of the main factors of the population dynamics in studied population. A significant genetic heterogeneity of HDs, both allelic and locus, was revealed in KChR.

## 1. Introduction

Genetic differentiation of human populations and its mechanisms remain the key problem of ethno-genetics and population genetics and became a clue for the understanding the intensity and direction of microevolution process in different ethnic groups. Investigation of genetic load and diversity of hereditary disorders (HDs) in human populations and its characteristics such as: size, structure (proportion of autosomal dominant (AD), autosomal recessive (AR) and X-linked (XL) disorders), temporal and geographical dynamics may be effective only if there is a close interaction between population and medical genetics study.

Unfortunately, our knowledge about the load and diversity of HDs and its peculiarities in different countries or ethnic groups is limited. In most cases, information on disease prevalence, could be found in the literature [1,2,3], which is based on the assessment of possible genetic risk in human populations exposed to ionizing radiation. A classic example is the Register of Congenital and Hereditary Pathology in Canadian Province British Columbia created in 1952, which gives the most accurate estimates of the frequency and structure of congenital and hereditary diseases: 1.4/1000 for AD, 1.7/1000 for AR and 0.5/1000 for XL disorders [4,5]. Unfortunately, information about the load and diversity of HDs obtained in particular ethnic groups could not be used for general considerations as well as for extrapolation to other populations. Load of accumulated specific and frequent HDs were identified in populations of Finns [6], Ashkenazi Jews [7], Amish from the USA [8], French Canadians [9,10] and some others. These studies demonstrated the role of the genetic structure and the influence of factors of population dynamics on the formation of the load and diseases specific to these populations. Works on ethnically different populations show the differentiation in the structure and diversity of the nosological spectrum of HDs. The global, predominantly European, prevalence of some nosological forms is currently covered in the annual report of the Orphanet Reports Series and in OMIM (Online Mendelian Inheritance in Man) [11].

However, it is also important to study modern and non-isolated human populations, to analyze how hereditary pathology is distributed in the region with a multi-ethnic composition, whether subpopulations have the accumulation of some hereditary diseases or whether there is a merge of gene pools as a result of population miscegenation (mixing of ethnic groups).

The aim of this publication is to study the load and diversity of HDs of the different populations of the Karachay-Cherkess Republic (KChR), to perform analysis of the causes of genetic variability and to carry out comparative analysis of the structure of HDs diversity in KChR with 12 previously surveyed populations of Russia. This can not only have significant impact on the local Health Care providing services but also uncover the population structure of modern open populations including its differentiation in pathogenic part of genome.

The structure of the ethnic composition of modern KChR in the North Caucasus has been formed over the past three thousand years. Representatives of more than 50 ethnic groups live on the territory of KChR but the major ones are 5 ethnic groups: Karachays, Russians, Circassians, Abazins and Nogais. Ethnogenesis of indigenous peoples of KChR is complex and not clearly defined. Karachays and Nogais belong to the Turkic-speaking peoples originated from different roots: Karachays are descendants of Alans-Koban ethnic group mixed with Turkic-speaking Bulgarians; Nogais originated from Turkic tribes Pecheneg and Kipchak. Circassians and Abazins belong to the group of Abkhazian-Adyghe peoples [12].

## 2. Results and Discussion

In Cherkessk city and 10 districts of the KChR, 8950 families (with 10,125 patients) with presumably the HDs were initially identified as a result of the survey and data collection through various sources of registration (from 1156 medical workers from 163 medical institutions). The analysis of our work has shown that the card questionnaire works best in rural populations, identifying more than 80% of all patients with the NB, as the medical staff is well informed about all patients at their site. The analysis of primary sources of registration in the entire population showed that the most informative data for patients with mental disorders, neurological and ophthalmological diseases were the data on disabled people, through which the largest number of patients were registered—67%, which is due to the centralized nature of storage of information about these groups of patients. Information on patients with orthopedic, skin diseases and hereditary syndromes was mainly provided by pediatricians (53–59% of patients), nurses (49% of patients) and specialized physicians (35%). Fifty-seven percent of patients are registered from two or more sources of information. Next, patients and their family members were examined and a cytogenetic study was carried out at the Cherkessk genetic consultation to exclude chromosomal pathology. Anamnesis was collected and pedigrees were analyzed. In our case such families turned out to be 3400.

After examination of all the patients and their relatives, a diagnosis of hereditary pathology was established in 1849 patients (from 1295 families) for whom medical records were drawn up containing full description of the clinical and genetic status, case history and the family pedigree. All families were subjected to the segregation analysis depending on the putative pattern of inheritance (autosomal dominant, autosomal recessive inheritance).

Segregation analysis. Virtually in all the populations studied, the segregation frequencies did not differ from the expected (0.25) segregation frequency for recessive diseases, with the proportions of sporadic cases being small. On average, the segregation frequency for a group of families with presumably AR disease inheritance was 0.22 ± 0.053 (mean ± SE), the probability of registration π = 0.86, the proportion of sporadic cases was 0.09 ± 0.008.

For the families wherein the hypothesis of the AD diseases inheritance was tested, we carried out two independent analyses (using the χ^2^ test and Weinberg’s proband method), depending on the method of the family ascertainment [13,14]. The segregation analysis performed in the families with registration through the affected parent showed that the ratio of the affected and apparently healthy people corresponded to the hypothesis of AD inheritance (50%). The segregation frequency calculated for a group of families with presumably autosomal dominant disease inheritance, registered through the affected children was 0.51 ± 0.091; registration probability π = 0.74.

Analysis of segregation frequencies for families with presumably AD or AR pathology did not reveal statistically significant differences from the expected value, which allowed us to proceed to the description of diversity and the calculation value of the genetic load of HDs. The use of multiple sources of registration gives a high percentage of detection of patients with NB, which follows from the values of the probability of registration. For families with AR type of inheritance, the probability of registration π = 0.86 is good in our studies but not complete. At the same time, for families with AD diseases, the obtained values of the probability of registration π = 0.74 indicate that 26% of patients may not have been detected by us. These results should be considered when calculating the load of the HDs.

### 2.1. The Structure of the Diversity of Hereditary Diseases in Populations of KCHR

The structure of the diversity of the HDs in accordance with the classification by organ and systemic types of the disease—Neurological, ophthalmic, genodermatoses, skeletal, hereditary syndromes and other pathology (hereditary diseases of metabolism, blood, hearing, etc.) is presented in Table 1 and in Appendix A. Most of the identified diseases were found in the previously examined Russian populations and countries of Europe [15,16,17,18].

Analysis of the structure of the nosological spectrum (Table 1) showed that there are 54 nosological forms (23.48%) of hereditary neurological diseases represented by 453 patients from 309 families (Appendix A). Most of the diseases were detected with a low occurrence, while 12 nosological forms were identified frequently (more often than 1:50,000): Charcot-Marie-Tooth disease 1A type—1:34,197 (average occurrence in Russia 1:13,785); Charcot-Marie-Tooth disease 2E type—1:31,567 (this form is found only in Karachays with occurrence 1:12,496); X-linked dominant Charcot-Marie-Tooth neuropathy—1:36,307 (in Russia 1:166,630); Duchenne muscular dystrophy type—1:41,137 men (in Russia—1:31,025 men); Huntington disease—1:36,307 (in Russia 1:60,655); neurofibromatosis type 1—1:36,307 (in Russia 1:16,663); early infantile epileptic encephalopathy type 4—1:41,037 (found only in Karachays with 1:16,244); undifferentiated mental retardation with AD inheritance—1:10,522 (in Russia 1:22,632), with AR inheritance—1:3085 (in Russia 1:9628) and with X-linked inheritance—1:2700 men (in Russia 1:10,800 men); Martin-Bell syndrome—1:13,679 men (in Russia 1:94,800 men); AR microcephaly with mental retardation—1:31,567 (in Russia 1:25,064). Among the Karachays, two nosological forms with a high occurrence were identified which were not found in the previously examined populations of Russia. Most hereditary neurological diseases are characterized by reduced fitness and longevity. The majority of families have applied for molecular diagnostics in order to plan further childbearing in the family.

A variety of hereditary ophthalmic diseases were diagnosed with 42 nosological forms in 204 patients from 158 families (Appendix A). Six nosological forms were diagnosed frequently—congenital hereditary cataract—1:12,070 (undifferentiated due to artiphakia in patients) and AD zonular cataract—1:31,567 (total occurrence of congenital cataract in Russia 1:15,014); hereditary congenital ptosis—1:27,358 (in Russia 1:25,273); keratoconus—1:41,037 (in Russia 1:303,000); optic nerve hypoplasia—1:41,037 (in Russia 1:94,700); Stargardt disease—1:24,129 (in Russia 1:178,000). This group of hereditary diseases is characterized by the adaptability of genotypes near 1, in most cases easily treatable (pathogenetically and surgically) and available for the rehabilitation of patients. Early identification, treatment and rehabilitation of patients allow to postpone the age of disability and achieve social adaptation of the majority of patients.

The group of hereditary genodermatoses (Appendix A) in the KChR presented 14 diseases (169 patients from 90 families). Five diseases were identified with the occurrence of more than 1:5000—ichthyosis vulgaris—1:13,679 (in Russia 1:5013); X-linked ichthyosis—1:14,156 men (in Russia 1:14,296 men); nonbullous form of AR ichthyosiform erythroderma—1:41,037 (in Russia 1:30,995); keratosis palmoplantaris striata I—1:5195 (in Russia 1:10,246); familial multiple lipomatosis—1:10,153 (in Russia 1: 30,946). Hereditary genodermatoses is the most favorable and adaptive group of the HDs. In patients with various forms of ichthyosis and hyperkeratosis palmoplantaris, clinical manifestations become less pronounced with age, patients are socially adapted and in most cases do not seek medical help in older age groups.

The spectrum of hereditary diseases of the musculoskeletal system consists of 29 nosologies which were revealed in 115 patients from 88 families (Appendix A). Four frequent diseases were: ectrodactyly—1:45,596 (in Russia 1:75,819, more often the disease occurred in Circassians 1:12,704); postaxial polydactyly—1:41,037 (in Russia 1:28,611, the occurrence in KChR Abazins is 1:8316); idiopathic scoliosis 1—1:45,596 (in Russia 1:46,658); osteogenesis imperfect—1:37,306 (in Russia 1:37,441). Patients in this group of HDs, in most cases, need surgical treatment and systematic outpatient observation.

The greatest number of nosological forms was revealed in the group ‘*Hereditary syndromes*’—62 diseases. This group is quite frequent among patients—23.43% (435 patients) (Appendix A). The prevalence of severe hereditary syndromes is much greater in children than in adults, which is associated with low adaptability and reduced longevity. Most syndromes were identified with a low occurrence, while only 3 syndromes were identified to be frequent (more often than 1:50,000)—Ehlers-Danlos syndrome—1:1205 (average in Russia—1:6724); Gilbert syndrome with clinical manifestations of the disease—1:29,312 (in Russia 1:165,000) with predominance in the Caucasian peoples (Karachays—1:27,074, Nogais—1:7371, Circassians—1:10,163); Aarskog-Scott syndrome—1:25,648, with accumulation in Abazins—1:2772 (in Russia 1:45,547). Syndromes such as Ehlers-Danlos and Gilbert are usually characterized by an improvement in the condition of patients with age and are found in all age groups.

The greatest number of patients was revealed in the group ‘Other hereditary pathology,’ which combines hereditary metabolic diseases, hereditary diseases of blood, hearing and so forth—only 29 diseases were revealed in 481 patients from 384 families (Appendix A). Nine diseases were revealed with an occurrence of more than 1:50,000—von Willebrand disease type 1—1:25,648 (in Russia 1:150,000), more often revealed in Karachays—1:10,830; hemophilia A—1:12,070 men (in Russia 1:18,694 men), hemophilia B—1:45,596 men (in Russia 1:75,819 men), G6PD-deficient hemolytic anemia (favism)—1:41,037 men (in Russia 1:303,000 men), more often revealed in Nogais 1:1474; oculocutaneous albinism, type IA—1:34,197 (in Russia 1:42,121); isolated growth hormone deficiency, type IA—1:45,596 (in Russia 1:43,953); autosomal recessive deafness type 1A—1:1840 (in Russia 1:4651); cystic fibrosis—the occurrence in the population of all age groups was 1: 17,842 (in Russia 1:104,577) more often revealed in Karachays 1:8550; phenylketonuria/hyperphenylalaninemia—the occurrence in the population of all age groups was 1:5130 (in Russia 1:75,819), accumulation was detected in the Karachays—1:2389.

### 2.2. Genetic Heterogeneity (Allelic, Locus) of Monogenic Hereditary Diseases in KCHR

Confirmatory molecular genetic diagnosis was performed for many nosological forms of both monogenic HDs and chromosomal diseases. In this section, we will dwell on vivid examples that demonstrate the presence of marked genetic heterogeneity (allelic, locus) that was detected in surveyed population.

The most striking example is phenylketonuria/hyperphenylalaninemia. Phenylketonuria (PKU)/Hyperphenylalaninemia (HPA) is diagnosed in 84 patients from 72 families. DNA confirmatory diagnosis was performed for the *PAH* gene. Material for molecular genetic analysis was obtained from 63 unrelated patients with serum phenylalanine concentrations several times higher than normal: Russians (*n* = 6), Circassians (*n* = 2), Abazins (n = 4), Nogais (*n* = 0), Karachays (*n* = 50). The spectrum of *PAH* mutations in Russians from KChR included nine different mutations: c.1222C>T, R408W (4/12 chromosomes); c.688G>A, V230I (1/12); c.1241A>G, Y414C (1/12); c.1066-11G>A (1/12); c.60+5G>T (1/12); c.664_665delGA, D222* (1/12); c.473G>A, R158Q (1/12); c.1045T>C, S349P (1/12); c.442-2913_509+1173del4154ins268, EX5delins (1/12). In the Circassians, four different *PAH* mutations were identified in two patients: c.1066-11G>A; c.1169A>G, E390G; c.1238G>C, R413P; and c.1089delG, Lys363Asnfs*37. In Abazins, seven different *PAH* mutations were detected in four patients: c.1238G>C, R413P (2/8); c.781C>T, R261X (1/8); c.143T>C, L48S (1/8); c.1222C>T, R408W (1/8); c.1243G>A, D415N (1/8); c.1208C>T, A403V (1/8); c.898G>T, A300S (1/8). The spectrum of the *PAH* of mutations in Karachays was as follows: c.781C>T, R261X (68/100); c.631C>A, P211T (10/100); c.1238G>C, R413P (6/100); c.688G>A, V230I (2/100); c.992T>C, F331S (2/100). Nine mutations were detected in a sample of patients on only one chromosome (c.1208C>T, A403V; c.1169A>G, E390G; c.1139C>T, T380M; c.506G>A, R169H; c.442-1G>A; c.158G>A, R53H; c.527G>A, R176Q; c.961C>A, L321I; c.631C>A, P211L). In three patients with HPA, only one allele is identified (3%). Identified 25 patients from KChR with HPA caused by *PAH* gene mutations (or at least one mutation) discussed in References [19,20]. According to newborn screening in the KChR, very high birth prevalence of HPA was determined, which is 1:850 newborns (PKU only—1:1581 newborns), 1:332 in Karachays, while the average prevalence at birth in Russia is 1:7000. Thus, birth prevalence of the disease was revealed to be the highest among registered in the world. The founder haplotype and mutation “age” were identified by the analysis of linkage disequilibrium between R261* and extragenic STR loci [20].

Cystic fibrosis (CF) was detected in 23 patients from 23 unrelated families. A number of patients died by the time of our examination. Molecular genetic analysis was performed in 15 unrelated CF patients: 14 Karachays and one Russian. The Russian CF patient has *CFTR* genotype F508del/2184insA. A high frequency (92.86%) of mutations W1282X (26/28 chromosomes) was determined in Karachays: twelve patients are homozygous for mutation W1282X, two others are compound heterozygous with second alleles R1066C and R709X. Thus, the most common in Russia F508del mutation in the *CFTR* gene was not found in Karachay patients with CF [15]. Screening of 341 healthy individuals (682 analyzed chromosomes) also showed absence of F508del mutation in population. Two mutations W1282X (6/682; 0.88%), 1677delTA (2/682; 0.29%) were detected. The birth prevalence of CF according to newborn screening in the KChR was determined to be high—1:2647 newborns (average birth prevalence in Russia is 1:11,585 newborns). Analysis of haplotypes of polymorphic DNA markers linked to *CFTR* (IVS1CA, IVS6aGATT, IVS8CA and IVS17bCA) showed that the origin of W1282X mutation in KChR and the Eastern European part of Russia was different. It is assumed that the penetration of the W1282X mutation in the Caucasus is associated with the migration of Jews from Iran in the early Middle Ages and its high frequency among CF patients in KChR is a consequence of the founder’s effect [21].

Hereditary non-syndromic sensorineural hearing loss (NSHL) revealed in 231 patients from 229 families. All patients were examined by a surdologist; most had profound prelingual sensorineural hearing loss. In two patients upon their statements, the debut of the disease was observed in 1–2 years old. Two families have a postlingual (with an onset up to seven years old) progressive deafness, while in the rest of the cases, deafness was congenital. Cochlear implantation was performed in two patients. DNA-diagnosis was performed in 127 patients with NSHL (50 Russians, 37 Karachays, 9 Circassians, 8 Abazins, 4 Nogais and 19 individuals of mixed or other ethnicity) from 91 unrelated families. Pathogenic variants of the *GJB2* gene were detected in 46 patients. In Russians, the allelic frequency of *GJB2* mutations among chromosomes of unrelated patients with deafness is 47.44%: 41.03% is the mutation c.35delG, 2.56% for each mutations c.313_326del14, c.-23+1G>A and c.269T>C. In Karachays mutant allele frequencies of c.35delG is 13.89%, c.358_360delGAG—2.78%; in Abazins: c.35delG—18.75% and c.235delC—6.25%; in the Circassians: c.35delG—31.25%, c.-23+1G>A—6.25%, c.101T>C—6.25%; for other ethnic groups: c.35delG—22.5%, c.358_360delGAG—5%, c.-23+1G>A—2.50%, c.224G>A—2.50%. In two Nogai brothers with progressive mixed type deafness, the pathogenic hemizygous nonsense substitution NM_000307.4 (*POU3F4*):c.907C>T was determined by the NGS sequencing and confirmed healthy carriage status in mother by Sanger sequencing. In a population of healthy Nogais females (198 chromosomes analyzed), mutation c.907C>T gene was not detected.

In 13 patients from one Karachay family with autosomal dominant form CMT, a novel heterozygous substitution in NM_006158.4 (*NEFL*):c.65C>A (p.Pro22His) was identified by NGS and confirmed in all affected family members by Sanger sequencing. Considering an intermediate median nerve conduction velocities in different patients fluctuated from 30 to 42 m/s and onset of the disease between 13 to 18 years with the appearance of weakness of peroneal group of muscles, autosomal dominant intermediate Charcot-Marie-Tooth disease type G is diagnosed.

Early infantile epileptic encephalopathy type 4 associated with heterozygous missense substitution NM_003165.3 (*STXBP1*):c.361T>C was detected in Karachay families.

In two families with congenital aniridia the DNA diagnosis by Sanger sequencing of the coding region of the *PAX6* gene and MLPA analysis of the 11p13 chromosome region revealed: in the first family (three patients with two sibs and their mother), a previously known pathogenic variant c.607C>T (p.Arg203Ter) in the heterozygous state, while in the second Karachay family with a sporadic case of congenital aniridia, a novel heterozygous deletion of five exons of the *PAX6* gene chr11:31778912_31794631del (hg18).

In the Circassian family with primary microcephaly and mental retardation type 5, NGS sequencing revealed a novel homozygous single nucleotide deletion NM_018136.4 (*ASPM*):c.1386delC leading to a preterm stop codon formation. A population analysis (202 chromosomes from healthy Circassians) shows that the population frequency of the c.1386delC variant is less than 0.005 [22].

In a Circassian family with a single patient with metatropic dysplasia, a novel de novo variant NM_021625.4 (*TRPV4*):c.245C>T in the heterozygous state was determined by NGS, which corresponds to the AD type of disease inheritance [23].

The above-mentioned examples of molecular genetic diagnosis show the presence of genetic heterogeneity, both allelic and locus, of monogenic HDs in KChR.

### 2.3. Analysis of Genetic Relationships for Prevalence of Hereditary Diseases between Districts and Main Ethnic Groups of KCHR

Based on the analysis of the diversity and prevalence of HDs we used cluster analysis to reveal genetic relationships between the regions under consideration. The result is shown in Figure 1. As follows from the dendrogram (Figure 1A), at an earlier level, the populations of the city of Cherkessk, Urupsky and Zelenchuksky districts unite into a single cluster. The Russian population is predominant in these districts (55–78%). At the next stage, the population of Ust-Dzhegutinsky district (represented by 22% Russians and 70% Karachays) joins and the three districts with predominant Karachay residents—Karachaevsky, Prikubansky and Malokarachayevsky (86%, 76% and 88%), respectively form a separate cluster at the level of 9.00. It is worth noting that the second ethnic group in the Malokarachayevsky district is Abazins (8%), which led to further clustering of the Abazinsky district (87% Abazyns). At the following steps, Khabezsky (96% Circassians), Adyge-Khablsky (multiethnic: 40% Circassians, 30% Abazins, 10% Russians, 6% Karachays) and Nogaisky districts (77% Nogais) join in succession. The results of cluster analysis of the HDs prevalence demonstrate genetic dependence on the ethnic composition of populations. This required further analysis to study the genetic relationships between the prevalence of HDs the main ethnic groups of the KChR.

The cluster analysis for the prevalence of HDs between the main ethnic groups of the KCHR shows (Figure 1B) the Abkhazo-Adyghean peoples demonstrate the greatest similarity to each other, while the Russian populations demonstrate the greatest differentiation. The data obtained in the cluster analysis show that in modern society, in the presence of high inter-population miscegenation (>20%), each ethnic group retains not only its traditions and language but also a significant part of the specific gene pool for each ethnic group.

### 2.4. Values of Hereditary Disease Load in Populations and Ethnic Groups of KChR

Table 2 shows the genetic load (per 1000 surveyed people) of main types of the HDs in the all populations of the KChR. An analysis of the genetic load in 11 populations of KChR showed the differences between the districts in terms of the load of AD, AR, X-linked pathology and total load of HDs (χ^2^_AD_ = 300.02; χ^2^_AR_ = 110.42; χ^2^_XL_ = 46.13; χ^2^_Tot_ = 402.82; d.f. = 10, *p* < 0.05). The total prevalence of HDs was 1:221 people (3.15 ± 0.145 among the Russian population, 5.55 ± 0.177 among Karachays, 5.55 ± 0.314 among Circassians, 5.22 ± 0.375 among Abazins, 7.51 ± 0.706 among Nogais). Differentiation is determined by comparative analysis of the load values of all considered groups HDs in different ethnic groups (χ^2^_AD_ = 127.7; χ^2^_AR_ = 112.74; χ^2^_XL_ = 36.45; χ^2^_Tot_ = 9.87; d.f. = 4, *p* < 0.05)—The lowest values were found in Russian population (3.15 ± 0.145), the largest—In Nogais (7.51 ± 0.706).

Hence, analysis of the load of HDs in the examined populations of KChR demonstrated the existence of a clear-cut differentiation between 11 studied populations. Moreover, there are strong grounds for believing that they also differ by the frequencies of genes of HDs. It should be noted that the resulting estimates of the load are similar to the indicators of load in other ethnic populations of Russia, by their absolute values are very close to the estimations of the HDs prevalence according to the data of the Register of Congenital and Hereditary Pathology in British Columbia (Canada).

However, we believe it is necessary to make some adjustments to the values of the estimated load. Taking into account the data of probability of registration determined for AR pathology (π = 0.86) and for families with AD pathology (π = 0.74), we can conclude that a certain number of patients could have been missed by us. This may be due to several reasons. The analysis of HDs load structure in patients of all age groups has shown that the majority of patients (51%) are children (under 17 years old), 31.91% of patients are in 18–45 years old group; while age group of 46 years old and older represents only 17.008%. Hereditary pathology in children is represented by more severe diseases. Most of these diseases are sharply reduced in fitness and are less common in older age groups. These diseases include neurological, psychiatric and hereditary syndromes associated with multisystemic involvement. Therefore, their prevalence in the whole population is determined by their prevalence in children. Also, the decrease in prevalence in the adult population may be associated with social adaptation and amelioration of clinical manifestations in some forms of genodermatoses, as well as with one of the most frequent hereditary syndrome—Ehlers-Danlos. Values of the total disease prevalence in the children’s population significantly prevail 1.5–3 times over the values of an adult population usually due to the severe HDs with early onset. Thus, the population prevalence is an age-dependent estimation. In addition, certain amendments should be made taking into account the limited number of nosological forms provided for in our study. However, it is necessary to take into account that the majority of newly detected diseases are represented by rare forms in specific families or populations and make up a small part of the HDs load in the whole population. Thus, it can be assumed that the HDs load in the surveyed population may be higher, mainly due to AD pathology and similar to the estimations by Carter et al., demonstrating load of AD diseases to be 7.0, AR—2.1 per 1000 population and XL—0.5 per 1000 men [11].

### 2.5. The Study of the Possible Causes of the Differentiation of KCHR Populations of Load of HDs

In order to elucidate which particular population genetic mechanisms are responsible for genetic differentiation of the populations of KChR by the load of HDs, the medical genetic study was simultaneously accompanied by the population genetic study. The genetic structure of the populations being surveyed was frequently described by means of the isonymic method, allowing the obtaining of virtually unbiased values of Wright’s random inbreeding (*F_st_*) for the population [24]. F_st_ values in populations were: Cherkessk City 0.00017, Ust-Dzhegutinsky 0.00192, Karachaevsky 0.00595, Malokarachaevsky 0.00558, Prikubansky 0.00309, Urupsky 0.00109, Zelenchuksky 0.00198, Abazinsky 0.00403, Khabezsky 0.0079, Adyge-Khablsky 0.00372 and Nogaisky 0.00547. High correlation coefficients were obtained between the values of AD, AR pathology load and the random inbreeding *F_st_* (*r* = 0.80 ± 0.200 and *r* = 0.69 ± 0.242, respectively). In the case of X-linked pathology, significant but lower correlation coefficient between load measures and *F_st_* was determined (*r* = 0.48 ± 0.202). The correlation coefficient between the total load (AD, AR and X-linked) and *F_st_* level was *r* = 0.78 ± 0.211. Since the values of *F_st_* constitute a measure of genetic drift, high values of correlation between random inbreeding and the load of hereditary diseases in the populations of KChR make it possible to suggest that genetic drift is probably one of the leading factors determining differentiation of the populations of KChR by the load of HDs. In this respect, KChR is similar to the rest of the populations of Russia and most of the populations of Europe [13,25,26,27].

## 3. Conclusions

Hence, the carried out studies have made it possible for the first time to determine the size of the load and spectrum of hereditary diseases characteristic of the populations of the Karachay-Cherkess Republic and for the main ethnic groups living in the territory. Frequent nosological forms of the HDs have been revealed, which are typical for each ethnic group as well as for the whole surveyed population. Analysis of the HDs diversity showed that 42 frequent diseases (with prevalence more than 1:50,000) accumulate the greatest number of patients in the population (*n* = 1380, 74.31%). The ratio was maintained for diseases with different types of inheritance: AD—68.34%, AR—80.36% and XL—81.63%. Diseases characteristic of specific ethnic groups were identified among the rare forms of the HDs. For a number of diseases (CF, PKU) high frequency of diseases is shown and allelic genetic heterogeneity is determined, which was a consequence of the founder. We also managed to demonstrate that one of the main factors of the population dynamics determining differentiation of the populations of Russia by the prevalence of Mendelian hereditary pathology, as well as determining local accumulation of hereditary diseases in the populations studied, is genetic drift. The presence of significant genetic heterogeneity was clearly displayed during cluster analysis. The study demonstrates the need to survey each modern population or ethnic group, because it expands our understanding of the spatial distribution and genetic heterogeneity of individual nosological forms by adding previously unknown genetic variants to the common knowledge about the pathological part of the human genome.

## 4. Materials and Methods

### 4.1. Survey Population

A medical and genetic examination of the population of the Karachay-Cherkess Republic (KChR) was carry out. The population attached to the health care providing institutions of the Republic was subject to inspection. Total size of population of KChR is 465,563 (2019). The size of surveyed population was 410,367 who received medical care in KChR, which is more than 88% of the Republic residents—Cherkessk City (138,900 people) and ten Districts—Ust-Dzhegutinsky (43,396), Karachayevsky (35,500), Malokarachayevsky (36,594), Prikubansky (27,557), Urupsky (18,074), Zelenchuksky (43,588), Abazinsky (14,631), Khabezsky (25,474), Adyge-Khablsky (11,178) and Nogaisky (15,475). Ethnically, the surveyed population of the Republic corresponds to the official administrative statistics: Karachays (162,444—40%), Russian (134,756—32%), Circassians (50,817—12%), Abazins (33,264—8%), Nogais (14,741—3.5%) and representatives of other ethnic groups (14,345—3.5%). The ethnic composition of Republic subpopulations is heterogeneous.: Karachays prevail (70–87%) in the Karachayevsky, Malokarachayevsky, Ust-Dzhegutinsky and Prikubansky districts, Russians (55–78%)—in the city Cherkessk, Urupsky and Zelenchuksky, Circassians—in Khabezsky (>90%), Abazins—in Abazinsky (>85%), Nogais—in Nogaisky (>80%). The Adyge-Khablsky district is multi-ethnic [12,28,29]. Sex ratio in the entire surveyed population: in pre-reproductive and reproductive groups—1:1 (M:F), in post-reproductive group—0.6:1 (M:F) [30].

### 4.2. Survey and Protocol

A survey of the investigated populations was conducted regardless of nationality, age and gender structure, in accordance with the protocol of genetic epidemiological studies (developed in the Research Centre for Medical Genetics, RCMG). The study design is based on the developed protocol of medical-genetic examination of small populations (development of RCMG). The protocol includes the study of populations through different genetic systems simultaneously: 1. study of the prevalence of hereditary diseases; 2. study of the genetic structure through the methods of population statistics; 3. study of the genetic structure through the DNA-loci of the nuclear genome. Methods of collection and processing of medical genetic material remain unchanged throughout all the studies conducted by the staff of RCMG and allow compare the newly obtained data with the results from the previously surveyed populations of Russia. 

The clinical and molecular genetic study was performed in accordance with the Declaration of Helsinki and approved by the Institutional Review Board of the Federal State Budgetary Institution “Research Center for Medical Genetics,” Moscow, Russia, (Protocol No. 5 dated 20 December 2010) with written informed consent obtained from each participant and/or their legal representative, as appropriate. Consent for publication was obtained from the legal guardians of the patients.

The protocol consists of the population genetic and medical genetic sections and is intended for comprehensive population analysis based on the data on different genetic systems, including the genes of HDs, DNA polymorphisms, demographic data including surnames frequencies [30]. Each of the studied districts of the region has one central district hospital, which includes in-patient and out-patient facilities and several rural outpatient clinics and hospitals in the villages. The number of rural medical institutions is determined by the number of villages in the region. Medical personnel permanently residing in the district have full information about the attached population. The size of actual population taking care in particular medical organization is determined by the Department of Medical Statistics of the hospital. The research is conducted in expeditions in three consecutive stages. At the first stage, a lecture course is conducted for all medical workers of the surveyed area (doctors of different specialties, nurses, paramedics) to explain the goals and objectives of the study. At the end of the lecture course, the medical personnel are given an Information card–questionnaire (Figure 2) with requirement for filling in the information on patients, including those at the initial stage of disease and with minimal clinical manifestations. The questionnaire contains easily detectable clinical symptoms of HDs, almost each of which is characteristic for a group of diseases (isolated, syndromic forms). This protocol allowed detection of maximum possible number of nosological forms of HDs known to date. The card–questionnaire with the listed symptoms allows to reveal practically all “portrait” syndromes.

Registration of families with presumably HDs was done through affected persons in the family. “Multiple registration” method was used [13]. In addition to the questionnaire card, other sources of information are involved—(i) Data on disabled persons from childhood (1–17 full years) and disabled adults (up to 48 years) in the district provided by the hospital; (ii) Rural outpatient clinics with data on patients in each village; (iii) Information from special schools for the blind and visually impaired, deaf and hard of hearing, boarding schools for children with intellectual disability; (iv) Data from the genetic consultation unit. Thus, registration of the same patient was possible from several sources of information. Information from all registration sources was recorded in a single database. Summing up personal data and additional sources allows to make a one-time “screening” of hereditary morbidity of the area. In the aggregate, the detection rate of patients from all sources of registration reaches 80%.

At the second step, the patients were examined by clinical geneticists from the RCMG. The patient with the presumed HD is given a medical card, which contains passport information of the proband and his family members, a brief medical history, genealogical data, detailed phenotyping data. If necessary, a cytogenetic study is conducted to rule out chromosomal abnormality. As a result of this work, a significant proportion of patients (usually more than half) from different sources is excluded from the sample because of the external cause of the disease (injuries, infections, etc.). The remainder is a list of families with presumably HDs for further research.

At the third step, clinical investigations were performed by the same qualified specialists from leading research institutes and Russian Ministry of Health (geneticist, neurologist, ophthalmologist, dermatologist, pediatricians, otolaryngologist and orthopedist), which ensured unification of diagnostic criteria. In some cases, molecular, cytogenetic, biochemical, X-ray, electromyography and other methods were used. The Protocol of investigation was approved by the Ethical Committee of the Research Centre for Medical Genetics. In all cases, patients and their relatives gave informed written consent to the examination, cytogenetic and molecular genetic analysis. Seventeen families refused to conduct the examination.

Since many HDs are heterogeneous, genetic analysis of the entire material was performed in order to test whether families with HDs had been correctly classified with respect to the mode of inheritance (autosomal dominant (AD), autosomal recessive (AR), X-linked (XL)) and to determine the absolute and relative numbers of sporadic cases [13,14].

A variety of HDs are presented in tables in the form of a registry in accordance with the classification by organ and systemic types of the disease: neurological, ophthalmic, genodermatoses, skeletal, hereditary syndromes and other pathology (hereditary diseases of metabolism, blood, hearing, etc.). The prevalence of diseases was calculated per 100,000 people. It should be noted that when describing the diversity of HDs, microdeletion, sporadic syndromes (Prader-Willi, Angelman, Beckwith-Wiedemann, etc.) with OMIM numbers, were not included in the load of HDs but used for the spectrum description. Comparative analysis of the diversity of HD was carried out for a number of Russian regions of the European part of Russia—seven Russian populations (Krasnodar krai, Kirov, Kostroma, Bryansk, Rostov, Tver regions, Karachay-Cherkess Republic), five ethnic groups of the Volga-Ural region (Mari people of Mari El Republic, Chuvash people of Chuvash Republic, Udmurt people of Udmurtia, Bashkirs of Republic of Bashkortostan, Tatars of Republic of Tatarstan) and Adygeans of Adygea, the total size of the population examined exceeds 3,250,000 people who represent 7 ethnic groups of the Russian Federation [7,8,9].

### 4.3. Molecular Genetic Analysis

Confirmatory DNA diagnostics was carried out in the laboratories of the RCMG: Laboratory of Genetic Epidemiology (head—R.A. Zinchenko), Laboratory of Epigenetics (head—Sci. V.V. Strelnikov), Laboratory of DNA Diagnostics (head—A.V. Polyakov). Variety of methods were used for DNA diagnosis—Sanger sequencing, MLPA, RFLP, AFLP, whole exome sequencing—Depending on the studied nosology according to the protocols published elsewhere by the authors of current manuscript [19,20,21,22,23].

### 4.4. Statistical Methods

Calculation of segregation frequency, probability of registration and proportion of sporadic cases was carried out by one of the methods used for multiple registration of families [13]. All values are presented as mean ± standard error (SE). The segregation frequency is calculated by the Weinberg proband method [31]. Probability of registration π is calculated by Fisher method [32]. In order to take into account the proportion of sporadic cases, a segregation analysis was carried out by the method of maximum likelihood taking into account the probability of registration in accordance with the algorithm of complex segregation analysis developed by Morton [33]. The prevalence and load of the NB population is calculated as the ratio of the real number of patients with a certain type of inheritance (AD, AR) to the number of surveyed population and normalized to 1000 people surveyed: f = *n*/(N/1000), where *n* is the number of patients, N is the number of surveyed population. The prevalence for X-linked pathology is calculated for 1000 men. Total load is calculated as the sum of AD, AR and XL load with the standard dispersion. For comparison of different populations by genetic load of hereditary pathology, χ^2^ distributions were used with the degrees of freedom (d.f.) equal to the number of groups compared minus 1.

### 4.5. Cluster Analysis

To study the genetic relationships between different populations/ethnic groups of KChR, we performed by cluster analysis through the median-joining method. The obtained matrix of genetic distances was used for building dendrograms by the nearest neighbors method in Statistica ver. 13 software. The variables used for cluster analysis were prevalence of each of 230 diseases per 100,000 people (the full spectrum and prevalence values could be found in Appendix A). The analysis was performed overall for AD, AR and X-linked diseases.

### 4.6. F_ST_ Analysis

Estimation of a random component of inbreeding *F_ST_* was obtained by the method of “isonymy” from the lists of voters. The frequencies of surnames characterized by selective neutrality and having informative value equal to the best codominant system were used as an indirect marker [34]. The surnames of the entire population of the KChR over 18 years of age were analyzed [35]. Calculation was carried out using the Crow and Manjah method [24].

### 4.7. Data Availability Statement

The datasets used and/or analyzed during the current study are available from the corresponding author on reasonable request.

## Figures and Tables

**Figure 1 ijms-21-00325-f001:**
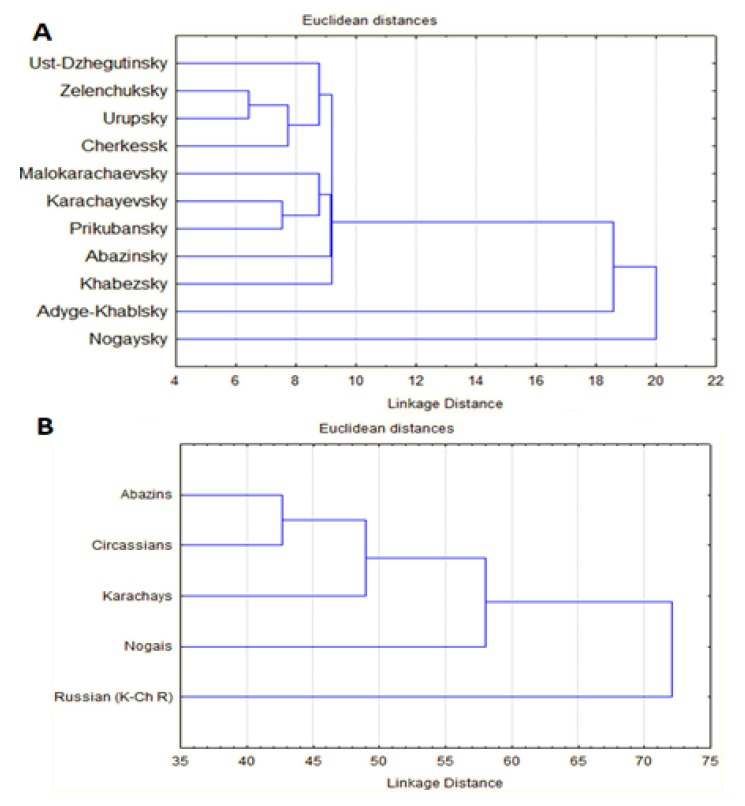
The genetic relationship between different districts of KChR (**A**) and between different populations/ethnic groups of Russia (**B**) by autosomal dominant (AD) and autosomal recessive (AR) diseases.

**Figure 2 ijms-21-00325-f002:**
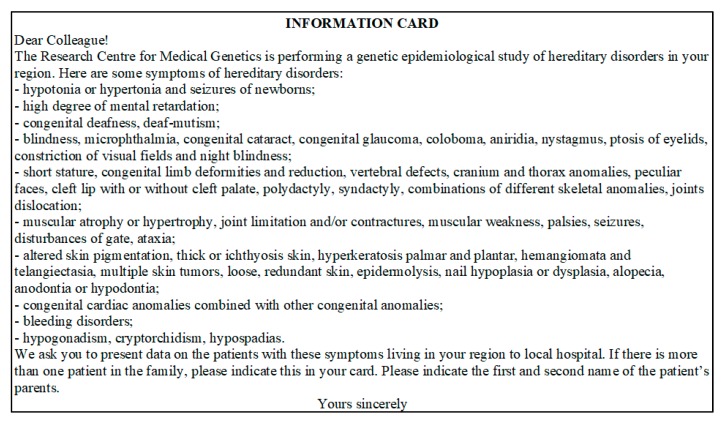
Information card–questionnaire.

**Table 1 ijms-21-00325-t001:** Structure of diversity and occurrence of hereditary diseases (HDs) divided by the main nosologic groups found in Karachay-Cherkess Republic (KChR) compared to the other populations of Russia.

Diagnosis	Number of Diseases (%)		Number of Patients (%)		Prevalence (per 100,000 People)	
	KChR	AvR	KChR	AvR	KChR	AvR
Neurological diseases	54 (23.48%)	87 (17.51%)	453 (24.39%)	1844 (23.59%)	110.39	66.50
Ophthalmic diseases	42 (18.26%)	66 (13.28%)	204 (10.99%)	1217 (23.59%)	49.71	43.89
Genodermatoses	14 (6.09%)	36 (7.24%)	169 (9.10%)	1229 (15.57%)	41.18	44.32
Skeletal diseases	29 (12.61%)	77 (15.72%)	115 (6.19%)	1000 (15.72%)	28.02	36.06
Hereditary syndromes	62 (26.95%)	191 (38.43%)	435 (23.43%)	1483 (12.79%)	106.00	53.48
Other hereditary diseases (diseases of metabolism, blood, etc.)	29 (12.61%)	40 (8.05%)	481 (25.90%)	1045 (13.37%)	117.21	37.68
Total	230	497	1857	7818	452.52	281.92

Note: AvR—Average for 12 populations of Russia.

**Table 2 ijms-21-00325-t002:** The genetic load of HD (per 1000 examined) in the populations of Karachay-Cherkess Republic (KChR).

District/City	Population	Load per 1000 People/Men *			
		AD	AR	XL *	Total
**Load in Administrative Units (Districts, City)**					
Ust-Dzhegutinsky District	43,396	2.30 ± 0.230	2.00 ± 0.215	0.74 ± 0.184	5.05 ± 0.328
Karachaevsky District	35,500	2.54 ± 0.267	2.06 ± 0.240	0.9 ± 0.225	5.49 ± 0.376
Malokarachaevsky District	36,594	3.96 ± 0.328	2.62 ± 0.267	0.66 ± 0.189	7.24 ± 0.433
Cherkessk City	138,900	1.03 ± 0.086	0.95 ± 0.083	0.36 ± 0.072	2.34 ± 0.125
Prikubansky District	27,557	2.58 ± 0.305	1.81 ± 0.256	1.16 ± 0.290	5.55 ± 0.424
Urupsky District	18,074	1.88 ± 0.322	1.27 ± 0.265	0.89 ± 0.312	4.04 ± 0.445
Zelenchuksky District	43,588	1.88 ± 0.208	1.65 ± 0.195	1.06 ± 0.220	4.59 ± 0.305
Abazinsky District	14,631	2.87 ± 0.442	1.71 ± 0.341	1.78 ± 0.492	6.36 ± 0.610
Khabezsky District	25,474	4.71 ± 0.429	2.67 ± 0.323	1.26 ± 0.314	8.64 ± 0.558
Adyge-Khablsky District	11,178	3.58 ± 0.565	3.4 ± 0.551	1.97 ± 0.593	8.95 ± 0.841
Nogaysky District	15,475	5.82 ± 0.611	2.58 ± 0.408	1.42 ± 0.428	9.82± 0.764
**Load in the Ethnic Groups**					
Karachays	162,444	2.68 ± 0.128	2.00 ± 0.110	0.86 ± 0.103	5.55 ± 0.177
Russian	134,756	1.43 ± 0.103	1.13 ± 0.091	0.59 ± 0.049	3.15 ± 0.145
Circassians	50,817	2.83 ± 0.236	1.69 ± 0.182	1.02 ± 0.201	5.55 ± 0.314
Abazins	33,264	2.35 ± 0.263	1.60 ± 0.229	1.27 ± 0.275	5.22 ± 0.375
Nogais	14,741	5.22 ± 0.594	2.10 ± 0.377	0.14 ± 0.136	7.51 ± 0.706
Other ethnos	14,345	2.37 ± 0.406	2.30 ± 0.3400	1.39 ± 0.441	6.06 ± 0.610
Total/average	410,367	2.33 ± 0.075	1.72 ± 0.065	0.81 ± 0.063	4.86 ± 0.104

Note: AD—Autosomal dominant type of inheritance, AR—Autosomal recessive type of inheritance, XL—X-linked type of inheritance. *—Load per 1000 men examined. Genetic load is presented as mean ± SE.

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
