# Peer review of "Epidemiology of Hereditary Diseases in the Karachay-Cherkess Republic"

_ijms, 2020, doi:10.3390/ijms21010325_

Round 1

Reviewer 1 Report

The paper describes epidemiology of hereditary diseases in the Karachay-Cherkess Republic.

While epidemiology and genetics aspects are correctly presented, is the paper appropriate for IJMS – maybe a genetics or medical journal are more suitable for the paper.

Objections:

1. Structure of the Matherials and Methos should be more pronunciated with subsections, e.g.

2.1 Survey population

2.2 Survey and protocols

etc.

2. Presentation of results

For example, the meaning of results should be explained: What does 0.22±0.05 mean – is it proportion and SE of proportion? This also applies to Table 4.2

3. Precision of results

For example, in expression 0.22±0.05 both parts have the same decimal precision. The part after ± should have one more decimal than part in front of ±.

4. Cluster analysis – this should be better explaines

- variables used for clustering

- algorithm for clustering

5. software used for analysis

Author Response

First of all, we would like to acknowledge Reviewers for their valuable comments and thorough analysis of the manuscript.

Reviewer 1

Comments and Suggestions for Authors

The paper describes epidemiology of hereditary diseases in the Karachay-Cherkess Republic. While epidemiology and genetics aspects are correctly presented, is the paper appropriate for IJMS – maybe a genetics or medical journal are more suitable for the paper.

Response: We have submitted the manuscript to the Special Issue "Medical Genetics, Genomics and Bioinformatics" of the International Journal of Molecular Sciences which collected papers on human genetics, genetics, and computational biology based on materials presented at “Centenary of Human Population Genetics” Conference 29–31 May, in Moscow. The submitted work was presented at this conference.

Objections:

Structure of the Matherials and Methos should be more pronunciated with subsections, e.g.

2.1 Survey population

2.2 Survey and protocols

etc.

Response: The text is changes accordingly.

Presentation of results

For example, the meaning of results should be explained: What does 0.22±0.05 mean – is it proportion and SE of proportion? This also applies to Table 4.2

Response: The text is changes accordingly. This is mean value (either frequency or genetic load) and SE.

Precision of results

For example, in expression 0.22±0.05 both parts have the same decimal precision. The part after ± should have one more decimal than part in front of ±.

Response: The results are changes accordingly.

Cluster analysis – this should be better explaines

- variables used for clustering

- algorithm for clustering

Response: We have expanded the description of Cluster analysis. The variables used for cluster analysis were prevalence of each of 230 diseases per 100000 people. Nearest neighbors method is used as algorithm.

software used for analysis

Response: We have expanded the description of Cluster analysis. The variables used for cluster analysis were prevalence of each of 230 diseases per 100000 people. Nearest neighbors method is used as algorithm. Statistica ver.13 software was used for statistical and cluster analysis.

Reviewer 2 Report

The work presented is very interesting but uneven. Some fragments are excessively lengthy and with few exceptions there is a complete lack of description of the methods used in molecular studies.  These exceptions appear in lines 394 (bidirectional sequencing of coding sequence) and 401 (sequencing). But this is not enough. It would be advisable to present the methods used more accurately (e.g. PCR, RT PCR - may be with sequence of primers, RFLP, next generetion sequencing, etc.). As cluster analysis also appears among statistical methods, maybe microarrays were also used. The molecular analysis was written in .... is not enough.

Author Response

First of all, we would like to acknowledge Reviewers for their valuable comments and thorough analysis of the manuscript.

Response: thank you for the comment. We have expanded the description of the methods to some extent. Unfortunately, we are unable to present all the details on molecular methods as there are a lot of them and this will lead to a strong increase in the size of the article. All methods with primers and protocols were published by the authors elsewhere as the work on genetic epidemiology and molecular genetics of particular disorders are conducted by the authors for decades. We listed references in appropriate places within the text as well as in the Materials and Methods section.

Round 2

Reviewer 1 Report

The paper is now much more readable:

Subtitles have improved the structure of the manuscript The results are better described, i.e.  statistical parameters are declared in text and tables. Description of cluster analysis is far better.